# Association of Single-Nucleotide Polymorphisms on *FURIN* and *EPHA2* Genes with the Risk and Prognosis of Undifferentiated Nasopharyngeal Cancer

**DOI:** 10.3390/ijms26178486

**Published:** 2025-09-01

**Authors:** Seddam Hares, Kamel Hamizi, Hamza Rahab, Maewa Hibatouallah Bounneche, Souhila Aouidane, Leila Mansoura, Manel Denni, Wissem Mallem, Ghania Belaaloui

**Affiliations:** 1Laboratory of Acquired and Constitutional Genetic Diseases (MAGECA), Faculty of Medicine, Batna 2 University, Batna 05000, Algeria; seddam.hares@gmail.com (S.H.); k.hamizi@univ-batna2.dz (K.H.); 2Faculty of Nature and Life Sciences, Batna 2 University, Batna 05000, Algeria; 3Faculty of Medicine, Batna 2 University, Batna 05000, Algeria; s.aouidane@univ-batna2.dz (S.A.); mansouraleila@yahoo.fr (L.M.); maneldenni@gmail.com (M.D.); mallemwissemdor@gmail.com (W.M.); 4Radiotherapy Department, Cancer Control Center (CLCC), Batna 05000, Algeria; 5National Biotechnology Research Center (CRBt), Constantine 25016, Algeria; h.rahab@crbt.dz (H.R.); h.bounneche@gmail.com (M.H.B.); 6Epidemiology and Preventive Medicine Department, Benflis Touhami University Hospital, Batna 05000, Algeria; 7Clinical Oncology Department, Cancer Control Center (CLCC), Batna 05000, Algeria

**Keywords:** undifferentiated nasopharyngeal cancer, *FURIN*, *EPHA2*, single-nucleotide polymorphism, risk, prognosis

## Abstract

The undifferentiated nasopharyngeal cancer (NPC) is a multifactorial disease mainly due to Epstein-Barr Virus (EBV) infection. The transmembrane tyrosine kinase ‘EphA2’ and the protease ‘Furin’ are implicated in the EBV entry into epithelial cells and other physiological processes. To gain insights into the association of single-nucleotide polymorphisms (SNPs) rs4702 and rs6603883 (*FURIN* and *EPHA2* genes, respectively) with the risk and prognosis of the NPC, the genotypes of 471 individuals (228 cases and 243 controls) were assessed alongside risk cofactors (sex, tobacco, alcohol, occupation, and recurrent Ear, Nose and Throat infections) and prognosis cofactors (Tumor stage, local invasion, lymph node involvement, and metastasis) using multivariable logistic regression. We found that only the rs4702 AG/GG genotypes were statistically significantly associated with a reduced risk of cancer, both in the overall population and in men (approximately 50% reduction). The rs4702 GG genotype was also associated with a low frequency of local tumor invasion in the whole population (OR = 0.382, *p* = 0.017, co-dominant model, and OR = 0.409, *p* = 0.02, recessive model), but heterozygous women were associated with a higher lymph node involvement (OR = 3.53, *p* = 0.031, co-dominant model, and OR = 3.62, *p* = 0.02, overdominant model). The rs6603883 GG genotype was associated, in the dominant model, with distant metastasis in the whole population (OR = 2.5, *p* = 0.024), with advanced clinical stage in men (OR = 2.22, *p* = 0.034), and with advanced clinical stage and distant metastasis in patients under 49 years (OR = 3.13, *p* = 0.009, and OR = 5.15, *p* = 0.011, respectively). Additionally, men having the rs6603883 GA genotype were associated with lymph node invasion (OR = 2.22, *p* = 0.027, overdominant model). Our study is the first to demonstrate that *FURIN* and *EPHA2* germline gene polymorphisms are associated with NPC risk (for rs4702) and prognosis (for both rs4702 and rs6603883), with sex-specific differences. These results need to be replicated and further investigated in other populations.

## 1. Introduction

Nasopharyngeal cancer (NPC), a squamous cell carcinoma of the upper aerodigestive tract, is a highly invasive multifactorial malignancy [1,2]. Its etiology, especially the undifferentiated form of NPC, is based mainly on Epstein Barr Virus (EBV) infection in endemic regions. However, the impact of environmental and dietary factors should not be omitted, as well as the existence of a favorable genetic background [3,4]. Although the EBV is ubiquitous worldwide, the geographical and ethnic distribution of NPC and the disease sex ratio are unexpectedly paradoxical. While the regions of Southeast Asia, Greenland, and Alaska represent high-risk endemic zones, this type of cancer is rare in Western countries. In Northern Africa, NPC is characterized by an incidence rate ranging from 0.33 cases in Egypt to 3.5 cases per 100,000 inhabitants in Algeria [5]. In this region, a bimodal incidence curve was observed, with the first peak observed in the adolescence-adulthood transition (15–24 years) and the second one in people aged 45 to 60 years [6]. Even though NPC is chemo- and radio-sensitive in 80% of non-metastatic patients, late-stage diagnosis, particularly when initial distant metastases or extensive tumor invasion are present, predicts a very poor prognosis and accounts for a significant number of deaths, thus representing a real public health problem [7].

EBV, a Herpesviridae infecting the global population, may be asymptomatic, cause infectious mononucleosis, or even malignancies like undifferentiated NPC [8,9,10,11,12]. EBV entry relies on envelope glycoproteins binding to specific cell membrane receptors, leading to cell immortalization [13]. EBV entry into epithelial cells begins with its attachment to the cell surface mediated by the interaction between viral gH/gL glycoproteins and epithelial cell integrins. Then, these glycoproteins, in addition to viral gB glycoprotein, bind to cellular receptors EphA2/NMHC-IIA and NRP1, respectively, to initiate membrane fusion. Finally, the gB glycoprotein changes its structural conformation from the pre-fusion form to the post-fusion form in order to trigger actual viral penetration [14,15,16,17,18]. The EBV gB glycoprotein contains a polybasic amino acid sequence Arg-X-Lys/Arg-Arg. This motif is the cleavage site of Furin, which is a constitutive cellular protease of the subtilisin type encoded by the *FURIN* gene. Its main role is to cleave all proteins containing the polybasic amino acid motif, enabling it to contribute to several physiological processes regulating the body’s homeostasis [19] or pathological processes, such as the activation of several pathogens, including SARS-CoV-2, influenza virus, and HIV [20,21,22,23]. Interestingly, the deletion of this motif from EBV gB glycoprotein reduces cell:cell fusion in epithelial and B cells [24]. The EphA2 is a transmembrane protein belonging to the receptor tyrosine kinase family encoded by the *EPHA2* gene. In addition to its crucial role in migration, angiogenesis, cell proliferation, and differentiation, it is considered a specific entry receptor for Kaposi’s sarcoma herpesvirus (KSHV) and EBV [15,16,17,18].

Genome-Wide Association Studies (GWAS) are powerful and hypothesis-free approaches aiming to identify genetic variations associated with disease susceptibility. As to the NPC, all the GWAS studies were conducted on Asian populations and identified most of the putative susceptibility single-nucleotide polymorphisms (SNPs) in the human leukocyte antigen (HLA) region [25,26,27,28,29]. However, GWAS have limits [30], and genetic associations may vary between populations. Hence, we opted to investigate the association of other SNPs located on genes encoding proteins with a strong biological implication in viral entry (EphA2 and Furin) and having a functional implication in the expression of these proteins. The selected SNPs are rs4702 (located in the 3′UTR region of *FURIN*) and rs6603883 (located in the promoter region of *EPHA2*) [31,32,33,34]. The examination of the “GWAS catalog” database (https://www.ebi.ac.uk/gwas/, accessed on 2 January 2025) for any associations of each of the chosen SNPs with NPC or for the GWAS studies regarding the NPC susceptibility or prognosis did not bring any reported association. Therefore, it remains unclear whether these polymorphisms may confer NPC susceptibility, especially in a North African population.

Our candidate-gene study was conducted to determine the associations between the SNPs rs4702 and rs6603883, located on the *FURIN* and *EPHA2* genes, respectively, with the risk and prognosis of NPC. To our knowledge, this is the first study to highlight the importance of studying such associations.

## 2. Results

### 2.1. Characteristics of the Study Population

As shown in Table 1, there is no significant difference between NPC cases and controls regarding the variables: age, sex, alcohol consumption, Body Mass Index (BMI), and occupational exposure. The sex ratio was 1.85:1. Some of these results were expected because both groups of participants were matched for age and sex. On the other hand, it was observed that tobacco consumption and recurrent ENT infections multiplied significantly the risk of NPC by 1.44 and 6.432, respectively. Furthermore, the majority of the patients were diagnosed at stages I, II, or III (65.4%); their tumor invasion did not reach bone or nerve structures (57.4%) and did not have initial metastases (86.4%). But at the same time, they had, at least, a bilateral adenopathy (63.6%).

### 2.2. Allele and Genotype Frequencies in the Study Population

Allele and genotype frequencies of the rs4702 (*FURIN*) and rs6603883 (*EPHA2*) are represented in Appendix A, respectively (Appendix A). Comparison of the allele frequencies in our control group with those in other populations from the Allele Frequency Aggregator (ALFA) project did not show a significant difference with the European population group (*p* = 0.72, *p* = 0.98, for the two SPNs, respectively), whereas the difference was significant with the Asian population group (*p* = 0.006, *p <* 0.0001, for the two SPNs, respectively) (Appendix A).

Both loci were in HWE in the control group (Appendix A). In contrast, only rs6603883 remained in HWE among cases, while rs4702 deviated from equilibrium, suggesting a possible association with disease risk [35].

### 2.3. Genetic Polymorphisms and NPC Risk

#### 2.3.1. NPC Risk in the Overall Population

The genotypic distributions of the two SNPs in NPC cases and controls are shown in Table 2. By using a univariate statistical analysis, we found that the rs4702 (*FURIN*) genotype is significantly associated with a lowering of the NPC risk by approximately 50% in three genetic models (for heterozygous in the codominant and the overdominant models and for genotypes with a G allele in the dominant model). These observations are consolidated after adjustment of the results according to tobacco consumption, occupational exposure, and the proportion of ENT infections (the unpaired variables between cases and controls having a *p*-value that is strictly less than 0.2; see Table 1).

On the other hand, the rs6603883 (*EPHA2*) did not show any association with NPC susceptibility even after adjustment.

Statistically significant associations between polymorphisms and NPC susceptibility in the overall population are represented in Figure 1.

#### 2.3.2. NPC Risk in the Gender-Stratified Population

As there are 1.85 times more men than women, we stratified the overall population following individuals’ gender in order to evaluate whether genotypes exhibited a gender-specific association with NPC risk (Table 3). We found that the association of r4702 (*FURIN*) genotypes with NPC risk lowering persists in heterozygous men (codominant and overdominant models) and in men harboring the G allele (in the dominant model), either in the univariate statistical analysis or after adjustment. As for the rs6603883 (*EPHA2*), it still did not show any association. Surprisingly, women’s NPC risk was not associated with these two polymorphisms.

**Table 3 ijms-26-08486-t003:** Genotype frequencies of rs4702 and rs6603883 polymorphisms and their association with NPC susceptibility after gender stratification.

Men		Cases ♂ (n = 148), n (%)	Controls ♂ (n = 158) n (%)	OR (95% CI)	*p*-Value	OR ^a^ (95% CI)	*p* Value ^a,#^
Genetic model	rs4702						
CoDominant	AA	69 (46.6)	44 (27.8)	1		1	
	GG	31 (20.9)	34 (21.5)	0.581 (0.314 to 1.077)	0.084	0.729 (0.375 to 1.416)	0.351
	AG	48 (32.4)	80 (50.6)	0.383 (0.227 to 0.644)	0.000	0.429 (0.245 to 0.725)	0.003 *
Dominant	AA	69 (46.6)	44 (27.8)				
	AG-GG	79 (53.4)	114 (72.2)	0.442 (0.275 to 0.710)	0.001	0.515 (0.309 to 0.858)	0.011 *
Recessive	AA-AG	117 (79.1)	124 (78.5)				
	GG	31 (20.9)	34 (21.5)	0.966 (0.558 to 1.672)	0.903	1.148 (0.634 to 2.079)	0.649
OverDominant	AA-GG	100 (67.6)	78 (49.4)				
	AG	48 (32.4)	80 (50.6)	0.468 (0.294 to 0.745)	0.001	0.483 (0.293 to 0.798)	0.005 *
	rs6603883				NS		NS
Women		Cases ♀ (n = 80), n (%)	Controls ♀ (n = 85) n (%)				
	rs4702				NS		NS
	rs6603883				NS		NS

OR: odds ratio, CI: confidence interval. NS: non-significant. ^a^. Variable(s) Adjusted with Occupational Exposure, Tobacco Consumption, and Repeated ENT Infection. * Statistically significant results. ^#^ The chi-square test was used in all comparisons.

Statistically significant associations between the polymorphisms and NPC susceptibility in the subgroups stratified by sex are represented in Figure 1.

#### 2.3.3. NPC Risk in the Age-Stratified Population

Age distribution of the overall population was evaluated using AIC calculation. The results showed that the unimodal distribution has a significantly lower AIC (−526.43) compared to the bimodal distribution (2052.72). Hence, and in order to test whether the observed genotypic associations are age influenced, we divided our population into two age groups (≤49 years and >49 years) (Table 4) according to the average age of cases and controls (as shown in Table 1). This stratification did not lead to any change in the associations observed in the overall population (Table 2).

**Table 4 ijms-26-08486-t004:** Genotype frequencies of rs4702 and rs6603883 polymorphisms and their association with NPC susceptibility after age stratification.

Age ≤ 49 Years		Cases (n = 116), n (%)	Controls (n = 117) n (%)	OR (95% CI)	*p*-Value	OR ^a^ (95% CI)	*p* Value ^a,#^
Genetic model	rs4702						
CoDominant	AA	52 (44.8)	35 (29.9)	1		1	
	GG	19 (16.4)	21 (17.9)	0.609 (0.286 to 1.295)	0.197	0.598 (0.266 to 1.344)	0.213
	AG	45 (38.8)	61 (52.1)	0.497 (0.279 to 0.883)	0.016	0.511 (0.275 to 0.951)	0.034 *
Dominant	AA	52 (44.8)	35 (29.9)				
	AG-GG	64 (55.2)	82 (70.1)	0.525 (0.306 to 0.901)	0.018	0.534 (0.299 to 0.954)	0.034 *
Recessive	AA-AG	97 (83.6)	96 (82.1)				
	GG	19 (16.4)	21 (17.9)	0.895 (0.453 to 1.771)	0.751	0.867 (0.418 to 1.801)	0.702
OverDominant	AA-GG	71 (61.2)	56 (47.9)				
	AG	45 (38.8)	61 (52.1)	0.582 (0.346 to 0.979)	0.041	0.604 (0.345 to 1.058)	0.078
	rs6603883				NS		NS
Age > 49 years		Cases (n = 112), n (%)	Controls (n = 126) n (%)				
Genetic model	rs4702						
CoDominant	AA	51 (45.5)	40 (31.7)	1		1	
	GG	24 (21.4)	27 (21.4)	0.697 (0.350 to 1.388)	0.304	0.629 (0.296 to 1.336)	0.228
	AG	37 (33)	59 (46.8)	0.492 (0.274 to 0.881)	0.016	0.459 (0.243 to 0.867)	0.016 *
Dominant	AA	51 (45.5)	40 (31.7)				
	AG-GG	61 (54.5)	86 (68.3)	0.556 (0.328 to 0.943)	0.029	0.512 (0.287 to 0.912)	0.023 *
Recessive	AA-AG	88 (78.6)	99 (78.6)				
	GG	24 (21.4)	27 (21.4)	1.000 (0.538 to 1.860)	1.000	0.936 (0.474 to 1.846)	0.848
OverDominant	AA-GG	75 (67)	67 (53.2)				
	AG	37 (33)	59 (46.8)	0.560 (0.331 to 0.949)	0.030	0.542 (0.305 to 0.963)	0.037 *
	rs6603883				NS		NS

OR: odds ratio, CI: confidence interval. NS: non-significant. ^a^. Variable(s) Adjusted with Occupational Exposure, Tobacco Consumption, and Repeated ENT Infection. * Statistically significant results. ^#^ The chi-square test was used in all comparisons.

Statistically significant associations between the polymorphisms and NPC susceptibility in the subgroups stratified by age are illustrated in Figure 1.

### 2.4. Genetic Polymorphisms and NPC Prognosis

#### 2.4.1. Prognosis of the Overall Population

We studied the association of rs4702 (*FURIN*) and rs6603883 (*EPHA2*) polymorphisms with the classical indicators of NPC prognosis: TNM stage, local tumor invasion (T), lymph node involvement (N), and the presence of metastasis (M). This was conducted for the different genetic models (Codominant, dominant, recessive, and overdominant). Only significant results are represented in Table 5.

We found that patients with the rs4702 GG genotype have a significantly lower risk (OR = 0.38 to 0.40) of developing a local tumor invasion reaching the bone and/or nervous structures compared to other patients. Alternatively, we can present this result as a 2.5 to 2.6 times likelihood for the AA genotype to be associated with an advanced local tumor invasion compared to the GG genotype. This was observed in the codominant model and the recessive model even after adjustment for sex, age, BMI, and Tobacco consumption (Table 5).

Additionally, the dominant genetic model for rs6603883 shows that patients with a GG genotype of this SNP are significantly associated with a more advanced TNM stage (OR = 1.76) and a greater metastatic capacity (OR = 2.35) than other patients with an AA or GA genotype. These associations persisted after adjustment, except for the TNM stage of the disease (Table 5).

To enhance clarity, significant associations of rs4701 with prognostic indicators in the overall population are visually summarized in Figure 2. The significant results of rs6603883 are represented in Figure 3.

#### 2.4.2. Prognosis of the Gender-Stratified Population

In order to evaluate whether genotypes exhibited a gender-specific association with NPC prognosis, we stratified the overall population following individuals’ gender (Table 6).

In men, the rs6603883 polymorphism has a significant association with the prognosis indicators of NPC so that men with the GG genotype are more likely to be diagnosed at an advanced stage (OR = 2.22) according to both codominant and dominant genetic models. Additionally, men who have a GG or GA genotype in the recessive model and those having a GA genotype in the overdominant model are more likely to have lymph node involvement (OR = 2.70 and OR = 2.22, respectively) (Table 6). On the other hand, the associations of the *FURIN* polymorphism (rs4702) did not reach statistical significance.

In a completely opposite situation, genotypic associations in women are observed only with the rs4702 polymorphism (Table 6). All women that are heterozygous for this SNP presented more aggressive lymph node invasion (in the dominant, codominant, or overdominant genetic models) with an OR up to (2.82, 4.14, and 3.92, respectively) in the univariate analysis. These associations maintained their significance in both the codominant and overdominant genetic models (OR = 3.53 and OR = 3.62, respectively) after adjustment for sex, age, BMI, and Tobacco consumption (Table 6).

To enhance clarity, significant associations of rs4701 and rs6603883 with prognostic indicators after subgroup stratification are visually summarized in Figure 2 and Figure 3, respectively.

#### 2.4.3. Prognosis of the Age-Stratified Population

In order to test whether the observed associations with NPC prognosis are age influenced, we divided the overall population into two age groups (≤49 years and >49 years) (Table 6), according to the average age of cases and controls (as shown in Table 1).

Despite the fact that the risk of poor clinical staging and distant metastases was about 5 times greater in patients with the GG genotype of rs6603883 who do not exceed 49 years of age (Table 6), this genotype showed no statistically significant effect in patients aged over 49. It is important to mention that all the results regarding the rs4702 association with prognostic parameters were not significant after this stratification.

**Table 6 ijms-26-08486-t006:** Genotype frequencies of rs4702 and rs6603883 polymorphisms and their association with NPC prognosis indicators after stratification following patients’ gender and age.

**Gender Stratification**								
**Parameters**	**Genetic Model**	**Genotype**	**Parameter Category n (%)**		**OR (95% CI)**	***p* Value ^#^**	**OR ^a^ (95% CI)**	***p* Value ^a^** ** ^,^ ** ** ^#^ **
♂	Genetic Model	rs6603883	I-II-III	IVA-IVB				
TNM stage (n = 148)	CoDominant	GG	25 (27.5)	26 (45.6)	1			
		AA	21 (23)	9 (15.8)	0.412 (0.159 to 1.070)	0.063	0.459 (0.169 to 1.249)	0.127
		GA	45 (49.5)	22 (38.6)	0.470 (0.222 to 0.994)	0.047	0.452 (0.206 to 0.992)	0.048 *
	Dominant	GG	25 (27.5)	26 (45.6)				
		GA-AA	66 (72.5)	31 (54.4)	0.452 (0.225 to 0.905)	0.025	0.454 (0.219 to 0.940)	0.034 *
♂		rs6603883	N0-N1	N2-N3				
Lymph node involvement (n = 148)	Recessive	GG-GA	39 (69.6)	79 (85.9)				
		AA	17 (30.4)	13 (14.1)	0.378 (0.167 to 0.855)	0.019	0.370 (0.157 to 0.872)	0.023 *
	OverDominant	AA-GG	37 (66.1)	44 (47.8)				
		GA	19 (33.9)	48 (52.2)	2.124 (1.068 to 4.227)	0.030	2.220 (1.094 to 4.507)	0.027 *
♀	Genetic Model	rs4702	N0-N1	N2-N3				
Lymph node involvement (n = 80)	CoDominant	AA	16 (59.3)	18 (34)	1			
		GG	5 (18.5)	7 (13.2)	1.244 (0.329 to 4.708)	0.747	0.914 (0.221 to 3.784)	0.901
		AG	6 (22.2)	28 (52.8)	4.148 (1.368 to 12.580)	0.009	3.53 (1.120 to 11.133)	0.031 *
	Dominant	AA	16 (59.3)	18 (34)				
		AG-GG	11 (40.7)	35 (66)	2.828 (1.088 to 7.352)	0.031	2.363 (0.869 to 6.426)	0.092
	OverDominant	AA-GG	21 (77.8)	25 (47.2)				
		AG	6 (22.2)	28 (52.8)	3.920 (1.364 to 11.263)	0.007	3.62 (1.224 to 10.693)	0.020 *
**Age Stratification**								
**Parameters**	**Genetic Model**	**Genotype**	**Parameter Category n (%)**	**OR (95% CI)**	***p* Value ^#^ **	**OR ^b^ (95% CI)**	***p* Value ^b^** ** ^,^ ** ** ^#^ **	
Age ≤ 49 years	Genetic Model	rs6603883	I-II-III	IVA-IVB				
TNM stage (n = 116)	CoDominant	GG	22 (30.1)	23 (53.5)	1			
		AA	12 (16.4)	3 (5.7)	0.239 (0.059 to 0.964)	0.041	0.198 (0.043 to 0.911)	0.038 *
		GA	39 (53.5)	17 (40.8)	0.417 (0.184 to 0.943)	0.034	0.355 (0.146 to 0.865)	0.023 *
	Dominant	GG	22 (30.1)	23 (53.5)				
		GA-AA	51 (69.9)	20 (46.5)	0.375 (0.172 to 0.819)	0.013	0.319 (0.136 to 0.749)	0.009 *
Age ≤ 49 years	Genetic Model	rs6603883	M0	M1				
Presence of metastasis (n = 116)	Dominant	GG	35 (34.3)	10 (71.4)				
		GA-AA	67 (65.7)	4 (28.6)	0.209 (0.061 to 0.715)	0.016	0.194 (0.055 to 0.682)	0.011 *

OR: odds ratio, CI: confidence interval. (^a^) Variable(s) Adjusted with BMI and Tobacco Consumption. (^b^) Variable(s) Adjusted with Sex, BMI, Tobacco Consumption, and Repeated ENT Infection. * Statistically significant results. ^#^ Comparisons were performed using the chi-square test, or Fisher’s exact test when frequencies were ≤5. Only significant results are represented.

## 3. Discussion

Despite the ubiquity of EBV infection, inter-individual clinical variability suggests that its oncogenic role in NPC is only observed in conjunction with other exogenous and/or endogenous factors [4,36]. Based on this holistic conception for NPC risk factors, our case-control study investigated the association of two SNPs that were not previously assessed (rs4702 (*FURIN*) and rs6603883 (*EPHA2*)) with the risk as well as the prognosis of NPC. Other cofactors were investigated likewise (sex, tobacco consumption, alcohol consumption, occupational exposure, and recurrent ENT infection). Overall, our genotyping results showed that the rs4702 is associated with the risk of NPC in the whole cohort, and this association persists only in men, while the rs6603883 does not show any association with this risk. On the other hand, the GG genotype of rs4702 is associated with a low frequency of local tumor invasion in the whole population and a higher lymph node involvement in heterozygous women, while the GG genotype of rs6603883 is associated with metastasis in the whole population and in men.

Before interpreting our genotyping results, we selected the cofactors that we found to be associated with NPC risk in a univariate analysis, and then we used these cofactors in a multivariate analysis with the genotypes. Among these cofactors, tobacco consumption and recurrent ENT infections were associated with NPC risk, which is consistent with other studies [4]. This association could be explained by the role of chronic inflammation in promoting nasopharyngeal carcinoma, irrespective of EBV infection [37].

Our results are in accordance with the known biological implications of both functional SNPs [31,32,33,34]. Indeed, it was shown that the disruption of Furin expression has a direct influence on the onset of viral infections (Influenza virus, HIV, or SARS-CoV-2) [22,23,31,38,39]. Additionally, on a genome-wide scale, the rs4702 G allele can alter Furin expression following miR-338-3p binding on the corresponding mRNAs, which hinders its translation, while the rs4702 A allele may escape the suppressive effect of this microRNA, leading to high Furin levels [32,40]. In line with this, our results show that individuals with the rs4702 AG genotype, alone or with individuals with the GG genotype, are more likely to have a lower risk of developing an NPC compared to individuals with the AA genotype (Table 2), and this association is persistent only in men but not in women. We can speculate that there may be a protective effect of the G allele through reduction of Furin expression. The persistence of such an association in men may be due to their higher susceptibility to viral infections compared to women, whose immunological and hormonal status offer significant protection against pathogens [41,42,43,44]. And having 1.85 times more men than women in our sample may explain the visibility of this association in all the samples.

Additionally, we found that individuals with the rs4702 GG genotype are less likely to present with an advanced local tumor invasion compared to individuals with the rs4702 AA genotype (Table 5). But after stratification of the data according to the gender, we unexpectedly found that women harboring the rs4702 G allele (AG or AG + GG) were more likely associated with the development of aggressive lymph node metastases compared to those with the rs4702 AA genotype, and this is consolidated in all the studied genetic models (Table 6). These observations suggest that the rs4702 G allele may increase the risk of lymph node metastases in women. There are two hypotheses for explaining the increase. The first one is the possible increase in Furin expression in these women. As the rs4702 G allele reduces Furin expression following miR-338-3p binding on the corresponding mRNAs [32], and estrogens down-regulate the level of miR-338-3p [45], it is possible that the inhibitory effect of miR-338-3p on Furin expression is diminished in women. This effect is even strengthened by the fact that nasopharyngeal tumor cells already have reduced levels of miR-338-3p compared to healthy cells [46]. Hence, the consequence of reduced levels of miR-338-3p would be an increase in Furin expression in women. Accordingly, it has been shown that increased Furin expression correlates with an increase in the processing of membrane matrix metalloproteinase type 1 (MT1-MMP), which is one of its substrates; the latter activates extracellular pro-gelatinase A (a major extracellular matrix-degrading enzyme), which positively regulates tumor growth, thereby increasing the aggressiveness of head and neck tumors [47,48,49]. Furin is also a key player in tumor invasion through the cleavage and activation of various proteins like the growth factors IGF and PDGF, which enhance cell proliferation, and the angiogenic factor VEGF, resulting in increased vascularization and tumor growth [50]. Moreover, it has been shown that the oncogene Plac1 promotes NPC cell proliferation, migration, and invasion via the Furin/NICD/PTEN pathway [51]. Alternatively, the second hypothesis for explaining the increase in tumor invasion in women would be the fact that the reduced level of miR-338-3p (secondary to its binding to the rs4702 G allele [32] and to the effect of estrogens [45]) is known to be positively associated with lymph node metastasis in many solid tumors [52,53,54], possibly through an upregulation of COA3, a mitochondrial transmembrane protein [54].

The other SNP of interest in our study was the rs6603883. Despite its involvement in the modification of the EphA2 expression [34], our results did not show any association between this SNP and the susceptibility to develop NPC in our population (Table 2). This observation persists even after stratifying the data by gender and age (Table 3 and Table 4). As the overexpression of EphA2 facilitates and promotes EBV infection in epithelial cells by initiating membrane fusion [15,16,17,18], this prompts us to suggest the presence of other candidate proteins that may influence the viral entry phase. This is supported by a recent work of Yinggui Yang et al., who found that the interferon-induced transmembrane protein-1 (IFITM1) can inhibit the interaction between EphA2 and viral glycoproteins gH/gL or gB via two residues, Tyr121 and Leu104, thus inhibiting EBV entry into epithelial cells even though they highly express EphA2 [55]. On the other hand, it is suggested that the EBV membrane fusion signal may be directly given by gB and NRP1 interaction and that the overexpression of NRP1 stimulates viral glycoprotein fusion activity [56].

Interestingly, our results showed that patients who are homozygous for the rs6603883 G allele are more likely to have distant metastasis (OR = 2.5), and this risk increases in patients under 49 years of age (OR = 5.15) (Table 5 and Table 6). This is consistent with the cumulative data in the literature. Indeed, Xiaoyin Ma et al. showed that the G allele of rs6603883 has a lower affinity to the transcription factor PAX2 compared to the A allele, resulting in a reduced EphA2 expression [34]. Also, the canonical intracellular signaling induced by EphA2 after binding to its natural ligand (Ephrin A1) plays a tumor-suppressive role in epithelial cells by inducing a cascade of reactions beyond the EphA2 (this is called the “forward effect” of EphA2/Ephrin A1 interaction). It induces the inhibition of Ras-PI3K-AKT and Ras-MAPK pathways, initiated by growth factors such as epidermal growth factor (EGF) and platelet-derived growth factor (PDGF), resulting in the inhibition of cell proliferation and migration [34,57,58,59,60,61,62]. Hence, we may explain our results by the reduction of EphA2 expression resulting in the downregulation of its immunosuppressive role. It is noteworthy that EphA2 has a dual function in cancers because it also has a tumor-promoting effect. This effect is due to the non-canonical signaling pathways, which are ligand-independent, and to the cascade of reactions beyond the Ephrin A1 (called the “reverse effect” of EphA2/Ephrin A1 interaction). This dual role of EphA2 depends, among others, on ligand availability. Despite this fact, EphA2 is nowadays an attractive target for therapy [60,63,64].

We also found consistent results in men who are homozygous for the G allele of rs6603883, as they are more likely to have an advanced clinical stage (OR = 2.22), and in men who have either the rs6603883 GG or GA genotype, as they are more likely to have lymph node invasion (OR = 2.70) (Table 6). These results may be explained by the influence of sex hormones. Indeed, Xing Wang’s team showed that male sex hormones increase phosphorylation of the Ser897 residue of EphA2 through activation of the androgen receptor/Src/RSK1 signaling cascade [65]. The phosphorylation of the Ser897 residue can be observed during the ligand-independent non-canonical signaling pathways of EphA2 [60,63,64]. Finally, we can notice that our results show associations of the rs6603883 GG genotype with worse prognosis indicators in young men (Table 6). The reason for this is unclear, but it has been shown that the existence of metachronous metastasis is more frequent in men and young patients with NPC [66].

Finally, we should mention that our findings were not previously reported by single SNP analysis-based GWAS studies [25,26,27,28,29]. Many possible explanations can be given for this apparent discordance. The first ones are inherent to GWAS characteristics: the too stringent thresholds, which can lead to missing SNPs with low statistical associations but strong biological relevance [30]; the absence of considering cofactors like smoking or ENT infections in most GWAS studies; and the use of the additive genetic model as a default. Secondly, it is important also to consider the differences in allele frequencies between populations because all the GWAS on NPC were conducted on Asian populations [25,26,27,28,29]. Interestingly, there was a significant difference in allele frequencies between our population (North African) and Asian populations (Appendix A). Given these variations in MAF and the pattern of linkage disequilibrium across ethnicities, genetic predisposition to NPC may differ between Asians and other ethnicities. Thirdly, there may be locus interactions between the studied SNPs, making them not detectable in single SNP analysis. In line with this, one GWAS study was interested in SNP-SNP interactions and succeeded in identifying a number of suggestive interaction loci that were missed by single SNP GWAS analysis. Interestingly, this study found that the rs4702 is interacting with the rs373521 located on the *APP* gene, which encodes for the amyloid-beta precursor protein (APP) [67]. This becomes even more insightful when we know that APP has significant effects on cancer cell proliferation, migration, and invasion, potentially contributing to tumor progression in cervical squamous cell carcinoma, as demonstrated recently using a novel biotechnology called Spatial Transcriptomics Sequencing [68].

Our study has several strengths enhancing its reliability, such as the methodology starting from the choice of biologically active genes and functional SNP, the use of an appropriate sample size, the matching of the cases and controls, and the use of logistic regression analysis in order to address concerns about potential confounding factors. Nevertheless, we acknowledge that our study has some limitations, such as the need for replication in other populations. In addition, candidate-gene studies inherently have the limitation of potentially missing other relevant genetic variants by focusing only on selected SNPs. Finally, while our associations are statistically significant, functional validation is still required to clarify the underlying biological mechanisms and causality.

In conclusion, our study showed novel associations of rs4702 and rs6603883 polymorphisms, located in the *FURIN* and *EPHA2* genes, respectively, with either NPC risk or prognosis. Regarding NPC risk, having the rs4702 AG or GG genotypes is associated with a lower risk of NPC compared to the AA genotype, and this observation persists only in men. The rs6603883 did not show any association with NPC risk. With regard to the prognosis, the rs4702 GG genotype is less associated with advanced local tumor invasion compared to the AA genotype. Whereas the rs6603883 GG genotype is mostly associated with distant metastasis in the whole population and with advanced clinical stage, and with lymph node invasion in men under 49 years. Additionally, having the rs6603883 GA genotype in men is more associated with lymph node invasion. These findings provide new insights into NPC genetic susceptibility and prognosis. Replicating this type of research in other populations and incorporating our results into future meta-analyses on nasopharyngeal carcinoma risk and prognosis factors are of undeniable necessity.

## 4. Materials and Methods

### 4.1. Study Population

This case-control study included a total of 471 unrelated participants: 228 cases suffering from NPC (Type III, according to the third WHO histopathological classification) who were diagnosed and confirmed clinically, radiologically, and by anatomopathological examination. All patients were recruited between 25 November 2019 and 7 January 2021, within the Radiotherapy department and the Clinical Oncology department of the Cancer Control Center (CLCC) in Batna, in the East of Algeria. Clinicopathological parameters and the Tumor-Node-Metastasis (TNM) classification were carried out during diagnosis and before treatment, following the eighth edition of the Union for International Cancer Control [69]. Patients presenting another histological type or a double localization were excluded. During the same period, we recruited 243 healthy controls without a personal and/or family history of tumor disease and matched them with the cases according to age (±5 years), sex, and place of residence.

The demographic information as well as professional exposure and toxic habits of all participants (Table 1) were recorded during a face-to-face interview that typically lasted around 10 min and began with the signing of a clear informed consent. A standardized questionnaire was used. It had been adapted so that most questions were binary (yes/no, presence/absence) in order to minimize ambiguity and facilitate statistical analysis. Each interview was conducted by one trained interviewer following the same procedure to ensure consistency. It is important to note that people who consume more than 20 packets of cigarettes and/or chewing tobacco in their lifetime are considered to be tobacco consumers. During the interview, the participants’ exposure to inhaled chemicals, wood or textile dusts, and fire fumes was classified as a risky occupational exposure, and a recurrent Ear, Nose, and Throat (ENT) infections (i.e., otitis and/or angina) was also considered if it generally occurred more than or equal to 4 times a year according to the participants’ declarations.

Precautions to protect participants’ information were implemented: anonymizing data to protect confidentiality, removing personal identifiers just before data analysis, and restricting access to the data. The study was approved by the Institutional Review Board of the National Biotechnology Research Center (CRBt) and conducted in accordance with the principles outlined in the World Medical Association’s Declaration of Helsinki and its subsequent amendments.

### 4.2. SNP Selection, DNA Extraction and Genotyping

The SNPs of interest, which are located on genes encoding proteins with a strong biological implication in viral entry (EphA2 and Furin), were selected from the NCBI SNP database (https://www.ncbi.nlm.nih.gov/snp/, accessed on 2 January 2025) based on the following criteria: SNPs located in the promoter region or the transcribed region of the *FURIN* and *EPHA2* genes (i.e., from the 3′-untranslated region to the 5′-untranslated region), (2) with a minor allele frequency (MAF) > 5% in a Caucasian population that is closer to the North African population, and (3) affecting the expression of the corresponding protein. As a result, two SNPs met these criteria (rs4702 and rs6603883, located on the *FURIN* and *EPHA2* genes, respectively).

Starting from whole blood samples frozen at −20 °C in EDTA tubes, genomic DNA was extracted using the PureLink^®^ Genomic DNA Mini Kit (Invitrogen^TM^, part of Thermo Fisher Scientific, Waltham, MA, USA) following the manufacturers’ instructions. Real-time PCR genotyping was performed to study the two polymorphisms, rs4702 and rs6603883, using the adequate TaqMan^®^ SNP Genotyping Assays (Applied Biosystems^TM^, part of Thermo Fisher Scientific, Waltham, MA, USA). Reactions were performed on the Applied Biosystems^TM^ 7500 Real-Time PCR System. Following the manufacturer’s program (Applied Biosystems™, TaqMan^®^ SNP Genotyping Assays Protocol, available at: https://assets.fishersci.com/TFS-Assets/LSG/manuals/MAN0009593_TaqManSNP_UG.pdf, accessed on 2 January 2025), the thermal cycling conditions involved: a Taq Polymerase activation at 95 °C for 10 min, followed by 40 cycles of successive denaturation at 95 °C for 15 s and elongation at 60 °C for 1 min.

### 4.3. Statistical Analysis

Based on the MAF of the two polymorphisms (available on the NCBI SNP database https://www.ncbi.nlm.nih.gov/snp/, accessed on 2 January 2025), we estimated the required sample size for this study by setting the study power at 80% and a genotypic relative risk equal to 1.5. For this purpose, we used the QUANTO program version 1.2.4, May 2009 (https://keck.usc.edu/biostatistics/software/, accessed on 2 January 2025), software dedicated to epidemiology-genetics studies. The resulting sample size required to achieve adequate power was 196 cases and 196 controls. The chi-square test was used to check the Hardy-Weinberg equilibrium (HWE) by comparing the difference between observed and expected allele frequencies in control cases for the both SNPs. With the same test, we performed a univariate analysis to assess the association between the various nominal variables (rs4702 and rs6603883 genotypes, sex, tobacco consumption, alcohol consumption, occupational exposure, recurrent ENT infection, and the two SNPs) and the risk and/or prognosis of NPC (we used Fisher’s exact test if the effectives were too small (less than 5) after stratification of the data). To better understand the strength of association of rs4702 and rs6603883 genotypes with the risk and/or prognosis of NPC, we adjusted the OR and the 95% confidence interval (CI) by performing a binary logistic regression measuring the effect of variables that demonstrated, in the univariate analysis, an association with a *p*-value that is strictly less than 0.2 [70]. Student’s *t*-test was used to compare the mean age of cases and controls. “To account for different possible modes of inheritance, each reflecting a distinct biological hypothesis of allele effect, we evaluated associations under the dominant model (tests if carrying at least one risk allele is sufficient to influence the phenotype), recessive model (tests if two copies are required), overdominant model (tests whether heterozygotes differ in risk compared with both homozygotes), and codominant model (evaluates each genotype separately).” When evaluating genotype association with prognosis indicators, the choice of the reference category of genotype was carried out to maintain consistency and homogeneity across the comparisons. It serves as a baseline for evaluating the relative risk of disease severity across other groups. A straightforward interpretation of the OR is given when the non-reference group has lower odds of experiencing the outcome. Conversely, the increased risk for the reference group can be inferred by inverting the OR. We used the Akaike Information Criterion (AIC) to evaluate the likelihood of unimodal and bimodal distributions of age [71]. All the results with a *p*-value that is less than 0.05 were considered as statistically significant. All statistical analyses were performed using SPSS 20.0 software (IBM, Chicago, IL, USA).

## Figures and Tables

**Figure 1 ijms-26-08486-f001:**
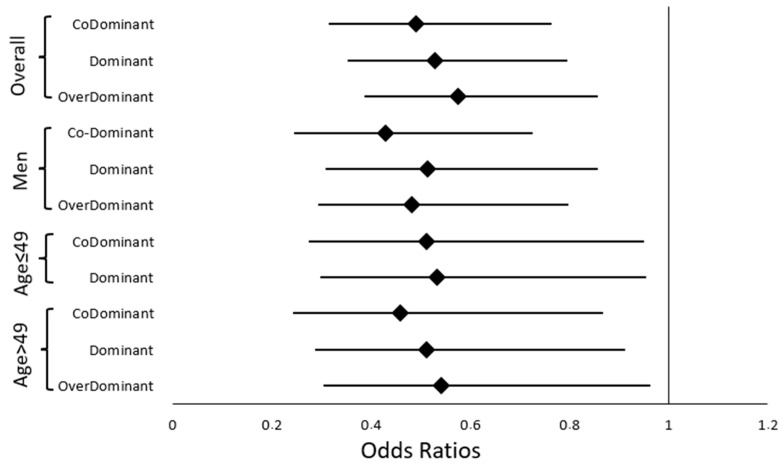
Forest plot summarizing significant associations between the rs4702 polymorphism and NPC susceptibility, as reported in Table 2, Table 3 and Table 4. The associations were observed in the overall population and, after stratification by age and sex, under multiple genetic models (co-dominant, dominant, recessive, and overdominant). Only significant results are shown. No significant association was observed for rs6603883.

**Figure 2 ijms-26-08486-f002:**
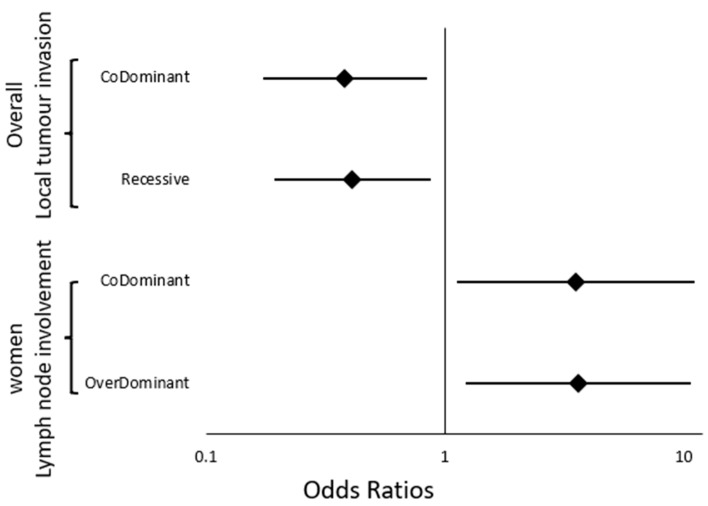
Forest plot showing the significant associations of rs4702 with prognostic indicators of NPC in the overall population and after stratification by sex, based on different genetic models. Results were extracted from Table 5 and Table 6. No significant results were found after age stratification for this SNP. A logarithmic scale was used for the odds ratio axis to accommodate variability in confidence intervals.

**Figure 3 ijms-26-08486-f003:**
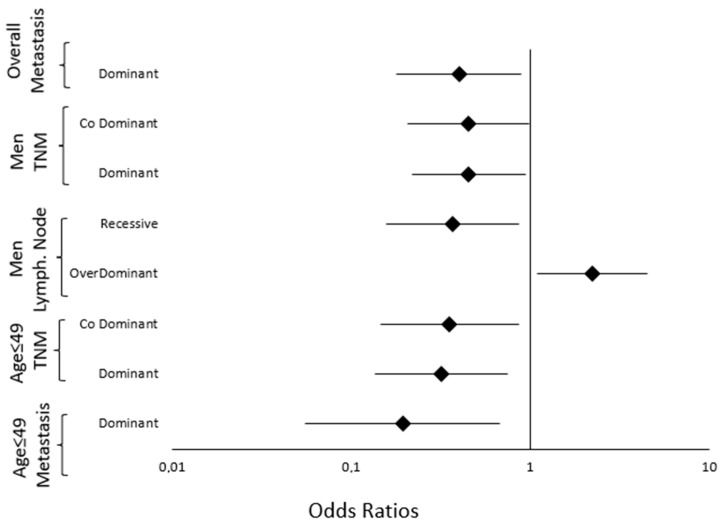
Forest plot illustrating the significant associations between rs6603883 and prognostic indicators of NPC. Associations are shown for the overall population and after stratification by sex and age, across various genetic models. Odds ratios (OR) with 95% confidence intervals are represented on a logarithmic scale to improve clarity. Data are based on Table 5 and Table 6.

**Table 1 ijms-26-08486-t001:** Description of the study population.

Variables	Cases (n = 228), n (%)	Controls (n = 243) n (%)	OR (95% CI)	*p* Value ^#^
Age (Mean ± SD, years)	49.21 ± 14.70	49.82 ± 14.68		0.65
Sex				
Male	148 (65)	158 (64.9)	1.00 (0.688 to 1.467)	0.98
Female	80 (35)	85 (35.1)		
Tobacco consumption				
Yes	123 (53.9)	109 (44.9)	1.440 (1.002 to 2.070)	0.048 *
No	105 (46.1)	134 (55.1)		
Alcohol consumption				
Yes	24 (10.5)	21 (8.6)	1.244 (0.672 to 2.302)	0.49
No	204 (89.5)	222 (91.4)		
BMI				
<18.5	8 (3.5)	7 (2.9)		0.93
18.5–25	88 (38.6)	95 (39.1)		
>25	132 (57.9)	141 (58)		
Occupational Exposure				
risky occupational exposure	134 (58.8)	124 (51)	1.368 (0.950 to 1.970)	0.09
non-risky occupational exposure	94 (41.2)	119 (49)		
Repeated ENT Infections				
Yes	89 (39)	22 (9.1)	6.432 (3.852 to 10.739)	0.000 *
No	139 (61)	221 (90.9)		
TNM stage				
I, II, and III	149 (65.4)			
IVA and IVB	79 (34.6)			
Local tumor invasion				
Ti, T1, and T2	131 (57.4)			
T3 and T4	97 (42.6)			
Lymph node involvement				
N0 and N1	83 (36.4)			
N2 and N3	145 (63.6)			
Presence of metastasis				
M0	197 (86.4)			
M1	31 (13.6)			

SD: standard deviation, BMI: Body Mass Index, ENT: Ear, Nose, and Throat. TNM: Tumor-Node-Metastasis, OR: odds ratio, CI: confidence interval. * Statistically significant results. ^#^ The chi-square test was used in all comparisons, except when comparing the mean age, where the Student’s *t*-test was applied.

**Table 2 ijms-26-08486-t002:** Genotype frequencies of rs4702 and rs6603883 polymorphisms and their association with NPC susceptibility.

Overall Population		Cases (n = 228), n (%)	Controls (n = 243), n (%)	OR (95% CI)	*p*-Value	OR ^a^ (95% CI)	*p* Value ^a,#^
Genetic model	rs4702						
CoDominant	AA	103 (45.2)	75 (30.9)	1			
	GG	43 (18.9)	48 (19.8)	0.652 (0.393 to 1.084)	0.099	0.625 (0.361 to 1.083)	0.094
	AG	82 (36)	120 (49.4)	0.498 (0.331 to 0.749)	0.001	0.491 (0.315 to 0.763)	0.002 *
Dominant	AA	103 (45.2)	75 (30.9)				
	AG-GG	125 (54.8)	168 (69.1)	0.542 (0.372 to 0.79)	0.001	0.529 (0.352 to 0.795)	0.002 *
Recessive	AA-AG	185 (81.1)	195 (80.2)				
	GG	43 (18.9)	48 (19.8)	0.944 (0.597 to 1.493)	0.806	0.914 (0.557 to 1.499)	0.72
OverDominant	AA-GG	146(64)	123 (50.6)				
	AG	82 (36)	120 (49.4)	0.576 (0.398 to 0.833)	0.003	0.576 (0.387 to 0.858)	0.007 *
	rs6603883						
CoDominant	GG	84 (36.8)	89 (36.6)	1			
	AA	40 (17.5)	39 (16)	1.087 (0.638 to 1.851)	0.76	1.354 (0.767 to 2.390)	0.29
	GA	104 (45.6)	115 (47.3)	0.958 (0.643 to 1.428)	0.834	0.970 (0.629 to 1.496)	0.89
Dominant	GG	84 (36.8)	89 (36.6)				
	GA-AA	144 (63.2)	154 (63.4)	0.991 (0.681 to 1.441)	0.961	1.062 (0.708 to 1.594)	0.77
Recessive	GG-AG	188 (82.5)	204 (84)				
	AA	40 (17.5)	39 (16)	1.113 (0.686 to 1.805)	0.664	1.378 (0.825 to 2.301)	0.22
OverDominant	GA-GG	124 (54.4)	128 (52.7)				
	AG	104 (45.6)	115 (47.3)	0.934 (0.65 to 1.341)	0.71	0.879 (0.594 to 1.300)	0.52

OR: odds ratio, CI: confidence interval. ^a^: Variable(s) Adjusted with Occupational Exposure, Tobacco Consumption, and Repeated ENT Infection. * Statistically significant results. ^#^ The chi-square test was used in all comparisons.

**Table 5 ijms-26-08486-t005:** Genotype frequencies of rs4702 and rs6603883 polymorphisms and their association with NPC prognosis indicators in the overall population.

Parameters	Genetic Model	Genotype	Parameter Category, n (%)		OR (95% CI)	*p*-Value	OR ^a^ (95% CI)	*p* Value ^a,#^
		rs4702	T1-T2	T3-T4				
Local tumor invasion (n = 228)	CoDominant	AA	53 (40.5)	50 (51.5)	1			
		GG	31 (23.7)	12 (12.4)	0.410 (0.190 to 0.886)	0.02	0.382 (0.173 to 0.843)	0.017 *
		AG	47 (35.8)	35 (36.1)	0.789 (0.440 to 1.415)	0.427	0.846 (0.467 to 1.535)	0.583
	Recessive	AA-AG	100 (76.3)	85 (87.6)				
		GG	31 (23.7)	12 (12.4)	0.455 (0.220 to 0.942)	0.028	0.409 (0.193 to 0.869)	0.020 *
		rs6603883	I-II-III	IVA-IVB				
TNM stage (n = 228)	Dominant	GG	48 (32.2)	36 (45.6)				
		GA-AA	101 (67.8)	43 (54.4)	0.568 (0.324 to 0.994)	0.048	0.578 (0.322 to 1.035)	0.065
		rs6603883	M0	M1				
Presence of metastasis (n = 228)	Dominant	GG	67 (34)	17 (54.8)				
		GA-AA	130 (66)	14 (45.2)	0.424 (0.197 to 0.913)	0.028	0.400 (0.180 to 0.888)	0.024 *

OR: odds ratio, CI: confidence interval. ^a^. Variable(s) Adjusted with Age, Sex, BMI, and Tobacco consumption. Only significant results are represented. * Statistically significant results. ^#^ The chi-square test was used in all comparisons.

## Data Availability

The new data underlying this article are available in the article.

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
