# Peer review of "Association of Single-Nucleotide Polymorphisms on FURIN and EPHA2 Genes with the Risk and Prognosis of Undifferentiated Nasopharyngeal Cancer"

_ijms, 2025, doi:10.3390/ijms26178486_

Round 1

Reviewer 1 Report

Comments and Suggestions for Authors

Seddam and colleagues studied the associates of two SNPs (rs4702 and rs6603883) with the risk and prognosis of NPC. The authors chose these SNPs, because they affect the expression of gene (protein level) implicated in viral entry, and a specific virus EBV, is linked to undifferentiated NPC. While the scientific results are easy to understand, this manuscript lacks impressive punchline except association with different factors. Furthermore, the flow of this manuscript needs improvement. Please find detailed comments below:

  1. The abstract lacks background and needs rewritten. Why these two SNPs are chosen. Also, it is very descriptive rather than a general abstract.
  2. The introduction spends two paragraphs in EBV, which most of them are un-related to this study. The authors need to think about how to structure the manuscript.
  3. 3. Key numbers and statistics should be shown in the main text so that readers don’t need to refer to Supplementary Tables. e.g. line116.
  4. This claim is confusing.

“Both loci were in HWE in the control group (Table 2S, supplementary tables). Whereas, in the group of cases, only the rs6603883 was in HWE (p > 0.05). Having the

rs4702 in HE disequilibrium is an indirect indicator of its potential association with the 123

disease”

Is rs4702 in HWE or not in the control and NCP groups?

  1. I am not an expert in genotype statistics, but the whole Table 2 is redundant. What is the point of showing Dominant, Recessive, and Over dominant numbers? They are just different combinations of varies alleles.

Author Response

Comment 1: The abstract lacks background and needs rewritten. Why these two SNPs are chosen. Also, it is very descriptive rather than a general abstract.

Response 1: We thank the reviewer for this valuable remark. We have revised the abstract to include a clearer background section, explaining the rationale behind the choice of the two SNPs. In addition, we have rewritten the text to ensure it is more concise. The revised parts have been highlighted in yellow in the updated version of the manuscript.

Comment 2: The introduction spends two paragraphs in EBV, which most of them are un-related to this study. The authors need to think about how to structure the manuscript.

Response 2: We thank the reviewer for this helpful comment. We agree that the introduction was overly detailed on EBV, which might have shifted the focus away from our main research question. We have restructured the section of the Introduction on EBV to include only information that highlights the biological roles of the proteins encoded by the studied genes, with particular emphasis on their involvement in the molecular mechanisms of EBV entry into epithelial cells and their functional relevance. On this basis, we focused on investigating SNPs within these genes, as their selection is biologically justified and aligned with the objectives of our study. The information presented also provides a solid foundation for the subsequent discussion of the results.

Comment 3: Key numbers and statistics should be shown in the main text so that readers don’t need to refer to Supplementary Tables. e.g. line116.

Response 3:

Thank you for your comment. We agree with the reviewer that key numbers should be visible in the main text. However, presenting the full allele and genotype frequencies in cases and controls for both SNPs would be bulky and distract from the main message. Instead, as the reviewer suggested, we have included the relevant p-values indicating differences between ethnic populations directly in the text, while keeping the detailed frequency data in the Supplementary Tables for clarity and accessibility.

Comment 4 : This claim is confusing.

“Both loci were in HWE in the control group (Table 2S, supplementary tables). Whereas, in the group of cases, only the rs6603883 was in HWE (p > 0.05). Having the rs4702 in HE disequilibrium is an indirect indicator of its potential association with the disease”

Is rs4702 in HWE or not in the control and NCP groups?

Response 4:

Thank you for your comment. The rs4702 is in HWE in the control group but not in the case groups, where it shows disequilibrium. This deviation suggests a potential association with disease risk, which was later confirmed by the genotype association analysis. We have rephrased the manuscript sentence to remove this ambiguity, as follows:

“Both loci were in HWE in the control group (Table 2S, supplementary tables). In contrast, only rs6603883 remained in HWE among cases, while rs4702 deviated from equilibrium, suggesting a possible association with disease risk”

Comment 5: I am not an expert in genotype statistics, but the whole Table 2 is redundant. What is the point of showing Dominant, Recessive, and Over dominant numbers? They are just different combinations of varies alleles.

Response 5 :

We thank the reviewer for this constructive comment. The inclusion of different genetic models is standard practice in association studies because they reflect distinct biological hypotheses of how a risk allele may act. For instance, in a dominant model, carrying at least one risk allele is sufficient to influence the phenotype, whereas in a recessive model, two copies are required. The overdominant model specifically tests whether heterozygotes differ in risk compared with both homozygotes, and the codominant model assumes a dose-response effect (it evaluates each genotype separately). Reporting these models ensures that no potential mode of inheritance is overlooked.

To address the reviewer’s concern, we have added an explanatory text in the Statistical Analysis section, to clarify this rationale.

“To account for different possible modes of inheritance, each reflecting a distinct biological hypothesis of allele effect, we evaluated associations under the dominant model (tests if carrying at least one risk allele is sufficient to influence the phenotype), recessive model (tests if two copies are required), overdominant model (tests whether heterozygotes differ in risk compared with both homozygotes), and codominant model (evaluates each genotype separately)”.

Reviewer 2 Report

Comments and Suggestions for Authors

The study's conclusions are fully consistent with its original objectives. The work provides new and potentially significant data on the genetic predisposition and prognosis of undifferentiated NPC in the North African population, with particular emphasis on gender differences. The need for replication in other populations is rightly identified as the next necessary step.

The article is well structured but contains minor typographical errors:

1. “SPNs” instead of “SNPs” in the abstract (p. 1).

2. “GAWS” instead of “GWAS” (p. 3, Introduction).

3. “s4702” instead of “rs4702” in the heading of Table 5 (p. 8).

4. “NCP” instead of “NPC” (p. 4, section 3.1).

Author Response

Comments 1:

The article is well structured but contains minor typographical errors:

  1. “SPNs” instead of “SNPs” in the abstract (p. 1).
  2. “GAWS” instead of “GWAS” (p. 3, Introduction).
  3. “s4702” instead of “rs4702” in the heading of Table 5 (p. 8).
  4. “NCP” instead of “NPC” (p. 4, section 3.1).

Response 1:

We thank the reviewer for pointing out the typographical errors. We have carefully revised the manuscript and corrected all such errors throughout the text as requested.

Reviewer 3 Report

Comments and Suggestions for Authors

I read with interest the manuscript entitled "Association of single-nucleotide polymorphisms on FURIN and EPHA2 genes with the risk and prognosis of undifferentiated nasopharyngeal cancer".

Identifying SNPs in FURIN and EPHA2 that correlate with NPC risk or prognosis could enhance understanding of the disease's molecular mechanisms, potentially leading to personalized treatment approaches.

The introduction is clear and concise, with a clear statement of the aim at the end.

Please clearly define how you sampled the controls. Are controls ensured to be representative of the same population at risk that produced the cases? If cases and controls are not adequately matched or randomly selected, biases may influence the findings.

You must provide all details related to the interview itself, including how it was conducted. How many researchers participated in the interview? How were the questions formatted? Etc.

When stating that you followed the manufacturer's instructions, please provide a reference to where the methodology has been described so far.

In tables, next to the p-value, indicate with a superscript which test was used.

Don't start sentences with "Figure 1..." and the like.

Please start the discussion section by presenting the most relevant results.

Please consider the limitations of the study further;

  • Focusing only on selected SNPs within FURIN and EPHA2 may overlook other relevant genetic variations influencing NPC risk or prognosis.
  • Identified associations may lack functional validation, making it difficult to infer biological mechanisms or causality.
  • Many genetic association studies are cross-sectional, limiting the ability to establish temporal or causal relationships between SNPs and disease progression.
  • Findings need replication in independent cohorts; lack of validation limits confidence in the results.

References are not written according to the instructions for authors. Please correct them.

Comments on the Quality of English Language

Please hire an English language proofreader. Many of the sentences are inadequate.

Author Response

Comment 1: Please clearly define how you sampled the controls. Are controls ensured to be representative of the same population at risk that produced the cases? If cases and controls are not adequately matched or randomly selected, biases may influence the findings.

Response 1:

We thank the reviewer for this remark.

We would like to clarify that the process of matching was already described in the Study population section, where we specified that “we recruited 243 healthy controls who do not have a personal and/or family history of tumor disease were homogenized with the cases according to age, sex and geographical location.” Please notice also that, in Section 1. Characteristics of the study population, we have already made an allusion to this matching by indicating that the comparison between cases and controls for age and sex was not significant, as expected, because of the matching procedure.

Accordantly, and to further avoid any ambiguity, we will highlight this aspect more clearly in the revised manuscript, Study population section (in yellow colour):

“We recruited 243 healthy controls without a personal and/or family history of tumor disease and matched them with the cases according to age (± 5 y), sex, and place of residence.”

Comment 2: You must provide all details related to the interview itself, including how it was conducted. How many researchers participated in the interview? How were the questions formatted? Etc.

Response 2:

We thank the reviewer for this valuable comment.

To answer your comment, we add this highlighted text (in yellow colour) in the Study population section:

The demographic information as well as professional exposure and toxic habits of all participants (Table 1) were recorded during a face-to-face interview that typically lasted around 10 minutes, and began with the signing of a clear informed consent. A standardized questionnaire was used. It had been adapted so that most questions were binary (yes/no, presence/absence) in order to minimize ambiguity and facilitate statistical analysis. Each interview was conducted by one trained interviewer following the same procedure to ensure consistency.”

Comment 3: When stating that you followed the manufacturer's instructions, please provide a reference to where the methodology has been described so far.

Response 3:

Thank you for your comment. We have now added a reference to the manufacturer’s program in the SNP selection, DNA extraction and genotyping section, as follows: “Following the manufacturer’s program (Applied Biosystems™, TaqMan® SNP Genotyping Assays Protocol, available at: https://assets.fishersci.com/TFS-Assets/LSG/manuals/MAN0009593_TaqManSNP_UG.pdf).”

Comment 4: In tables, next to the p-value, indicate with a superscript which test was used.

Response 4:

Thank you for your comment.

A superscript has been added next to the heading of the p-value column to indicate the statistical test used.

  • In Table 1, we added the following note: “# The chi-square test was used in all comparisons, except when comparing the mean age, where the Student’s t-test was applied.”
  • In Table 6, we added the following note: # Comparisons were performed using the chi-square test, or Fisher’s exact test when frequencies were ≤5.
  • In the other tables, we added the following note: “# The chi-square test was used in all comparisons.”

Comment 5: Don't start sentences with "Figure 1..." and the like.

Response 5:

We thank the reviewer for this stylistic suggestion. We have revised the sentence to avoid beginning with a figure reference. The sentence (in page 7) now reads: Statistically significant associations between the polymorphisms and NPC susceptibility in the subgroups stratified by age are illustrated in Figure 1.”

Comment 6: Please start the discussion section by presenting the most relevant results.

Response 6:

We thank the reviewer for this valuable suggestion. We have revised the beginning of the Discussion to make it more concise and now introduce it by highlighting the most relevant results. A paragraph presenting these key findings has been placed at the start of the section. These modifications are highlighted (in yellow colour) in the revised text.

Comment 7: Please consider the limitations of the study further;

  • Focusing only on selected SNPs within FURIN and EPHA2 may overlook other relevant genetic variations influencing NPC risk or prognosis.
  • Identified associations may lack functional validation, making it difficult to infer biological mechanisms or causality.
  • Many genetic association studies are cross-sectional, limiting the ability to establish temporal or causal relationships between SNPs and disease progression.
  • Findings need replication in independent cohorts; lack of validation limits confidence in the results.

Response 7:

We appreciate the reviewer’s insightful comment.

We acknowledge the limitations raised. Some of these are indeed imputable to the design of case-control studies, which are widely used in genetic association research. However, it is important to note that our work is not a cross-sectional study but a case-control analysis, which provides stronger evidence regarding associations with disease risk. However, we added your suggestions in the limitation section as highlighted (in yellow colour) in the text:

“Nevertheless, we acknowledge that our study has some limitations, such as the need for replication in other populations. In addition, candidate-gene studies inherently have the limitation of potentially missing other relevant genetic variants by focusing only on selected SNPs. Finally, while our associations are statistically significant, functional validation is still required to clarify the underlying biological mechanisms and causality”.

Comment 8: References are not written according to the instructions for authors. Please correct them.

Response 8:

We thank the reviewer for this remark. The references have been written and formatted according to the MDPI style guidelines.

Comments on the Quality of English Language

Please hire an English language proofreader. Many of the sentences are inadequate.

We thank the reviewer for this remark. We have carefully revised the manuscript and reformulated the inadequate sentences to improve clarity, readability, and overall English quality. These modifications are highlighted in blue in the revised version.

Reviewer 4 Report

Comments and Suggestions for Authors

This is an interesting retrospective study looking for possible associations among two single-nucleotide polymorphisms (SPNs) in FURIN and EPHA2 genes (rs4702 and rs6603883 respectively) with the risk and prognosis of the undifferentiated nasopharyngeal cancer (NPC. Statistical methods, number of cases and personal and clinical data of patients and controls are correct. ). Genome-Wide Association Studies was used for the SPNs detection.

I only have the following minor points/suggestions for addressing before definitive acceptance.

  1. I could not find supp. Tables. These tables should be available.
  2. The title seems to be an apparent proposal indicating the association with the SPN in both genes, but the results do not prove the association with EPHA2 gene. The abstract is not clear concerning a clear conclusion, and the final conclusion section is missing. Discussion is verbose to probably too long. In sum, I feel that a clear conclusion is needed even it is assumed that results need to be replicated and corroborated in further investigations on other populations.
  3. Introduction and mostly the discussion with association between NPC and EBV is probably out of the scope unless a link would be inferred from the data. I could not find this link. If so, please a clear emphasis would be included. Otherwise, this introduction would be shortened to focus the study.

Author Response

Comment 1: I could not find supp. Tables. These tables should be available.

Response 1:

Thank you for your comment. The supplementary tables were already uploaded during the initial submission as a zipped folder (.rar). Please let us know if there is a problem.

Comment 2: The title seems to be an apparent proposal indicating the association with the SPN in both genes, but the results do not prove the association with EPHA2 gene. The abstract is not clear concerning a clear conclusion, and the final conclusion section is missing. Discussion is verbose to probably too long. In sum, I feel that a clear conclusion is needed even it is assumed that results need to be replicated and corroborated in further investigations on other populations.

Response 2:

We thank the reviewer for all these remarks.

Regarding the title, our results show that rs4702 (on the FURIN gene) is significantly associated with both the risk and prognosis of NPC, whereas rs6603883 (on the EPHA2 gene) is associated only with the prognosis, but not with the risk. Accordingly, our title aimed to reflect both aspects: the involvement of SNPs in risk and/or prognosis, depending on the gene. But the distinctions are further clarified in the abstract, results, and discussion. We would have formulated the title as “…with the risk and/or prognosis of undifferentiated nasopharyngeal cancer”; however, this expression is neither precise nor commonly used in titles. For this reason, we decided to retain our initial formulation.

Regarding the abstract, we would like to clarify that it already includes a concluding statement summarizing the main findings but we reformulated it to be more clear: “…Our study is the first to demonstrate that FURIN and EPHA2 germline gene polymor-phisms are associated to NPC risk (for rs4702) and prognosis (for both rs4702 and rs6603883), with sex-specific differences. These results need to be replicated and fur-ther investigated in other populations”.As the abstract should remain concise, we cannot provide further details there, since the necessary precisions are already developed in the main text of the abstract.

In addition, the manuscript does contain a dedicated conclusion section, which starts with “In conclusion, …” and provides a synthesis of the study outcomes.

Concerning the shortening (conciseness) of the discussion, we have revised the text to make it more concise while improving the English. The modifications are highlighted in the revised manuscript

Comment 3: Introduction and mostly the discussion with association between NPC and EBV is probably out of the scope unless a link would be inferred from the data. I could not find this link. If so, please a clear emphasis would be included. Otherwise, this introduction would be shortened to focus the study.

Response 3: 

We thank the reviewer for this valuable comment.

The aim of the information provided in the introduction was first to highlight the biological role of the proteins encoded by the studied genes, emphasizing their implication in the molecular mechanisms of EBV entry into epithelial cells and their functional relevance. Based on this, we chose to investigate SNPs located in these genes, as their selection appeared biologically justified and consistent with the objectives of our study.

In addition, the introduction presented the state of association of these SNPs in previous GWAS, which we considered essential to further support our choice.

Finally, the information included in the introduction also provides a solid basis for the subsequent discussion of the results.

Accordingly, we have revised the introduction to focus more clearly on the biological and functional justification of the SNPs included in our study, while shortening content that may appear tangential.

Round 2

Reviewer 1 Report

Comments and Suggestions for Authors

The authors addressed my comments.

Author Response

Comment: The authors addressed my comments.

Response: We thank the reviewer for acknowledging that our revisions have addressed the previous comments.

Reviewer 3 Report

Comments and Suggestions for Authors

When citing the term "standardized questionnaire", please add a reference if it has been used previously.

When I think of manufacturer's instructions, I mean a reference to an article where the procedure is previously described in detail.

Author Response

Comment 1: When citing the term "standardized questionnaire", please add a reference if it has been used previously.

Response 1:

We appreciate your comment. Yes, you are correct that the questionnaire used in this study was specifically developed by our research team for the purposes of this project and has not been previously published or validated elsewhere. To avoid confusion, we have revised the manuscript to clarify that this was a “structured questionnaire” rather than referring to it as a “standardized questionnaire.”

Comment 2: When I think of manufacturer's instructions, I mean a reference to an article where the procedure is previously described in detail.

Response 2:

We thank the reviewer for this comment. In this study, we strictly followed the protocol provided by the manufacturer (ThermoFisher Scientific) without modifications. To the best of our knowledge, this procedure has not been described in detail in a peer-reviewed article. In other published studies using a similar procedure, the authors also refer directly to the manufacturer’s instructions without providing further methodological details. Therefore, we have cited the official manufacturer’s manual available on the ThermoFisher website, which represents the primary source of the instructions we followed.